# Suppression of the Electrical Crosstalk of Planar-Type High-Density InGaAs Detectors with a Guard Hole

**DOI:** 10.3390/mi13101797

**Published:** 2022-10-21

**Authors:** Jiaxin Zhang, Wei Wang, Haifeng Ye, Runyu Huang, Zepeng Hou, Chen Liu, Weilin Zhao, Yunxue Li, Xu Ma, Yanli Shi

**Affiliations:** 1Key Lab of Quantum Information of Yunnan Province, School of Physics and Astronomy, Yunnan University, Kunming 650504, China; 2Shanxi Guohui Optoelectronic Technology Co., Ltd., Taiyuan 030006, China

**Keywords:** InGaAs FPAs, electrical crosstalk, guard-hole structure, guard-ring structure, inoperable pixel

## Abstract

The resolution of InGaAs FPA detectors is degraded by the electrical crosstalk, which is especially severe in high–density FPAs. We propose a guard-hole structure to suppress the electrical crosstalk in a planar-type 640 × 512 15 μm InGaAs short wavelength infrared FPA detector. For comparison, the frequently used guard ring is also prepared according to the same processing. The calculation results show that the electrical crosstalk with a guard hole is suppressed from 13.4% to 4.5%, reducing by 66%, while the electrical crosstalk with a guard ring is suppressed to 0.4%. Furthermore, we discuss the effects of the guard ring and the guard hole on the dark current, quantum efficiency, and detectivity. Experimental results show the detector with a guard-hole structure has higher performance compared with the detector with a guard-ring structure, the dark current density is reduced by 60%, the QE is increased by 64.5%, and the detectivity is increased by 1.36 times, respectively. The guard-hole structure provides a novel suppression method for the electrical crosstalk of high-density InGaAs detectors.

## 1. Introduction

InGaAs short wavelength infrared (SWIR) focal plane array (FPA) detectors have attracted extensive attention due to their high detectivity, quantum efficiency, low dark current, and good anti-radiation characteristics in uncooled conditions. In addition, InGaAs FPA detectors have wide applications in aviation safety, biomedicine, camouflage recognition, night vision, and other fields [1,2]. With the reduction in the pitch and the increase in the format, the electrical crosstalk of InGaAs FPA detectors has become an important problem affecting the resolution of detectors, especially for those applications that rely on quantitative image information in image data and accurate image correction [3], which puts forward higher requirements for reducing the electrical crosstalk of InGaAs detectors.

Recently, studies on the electrical crosstalk of InGaAs FPAs have been carried out using different methods [4,5,6,7]. Estribeau M. et al. masked the active area to reduce crosstalk by recovering a symmetry without an important loss in quantum efficiency of the CMOS imager sensor with Modulation Transfer Function (MTF) [8]. Dongxue Li et al. studied the electrical crosstalk of typical planar and mesa InGaAs/InP as a function of the illumination wavelength and incident angle, as well as the etching depth in the mesa structure with numerical simulation. The results showed that the back-side illumination had higher electrical crosstalk than that of the front-side illumination, and the electrical crosstalk of the mesa structures was better than that of the planar structure at any wavelength [9]. Xiumei Shao et al. studied a planar-type 24 × 1 InGaAs detector with a guard ring using the sealed-ampoule diffusion method, and the results demonstrated that the sensitive area of the pixels was effectively restricted within the designed area by the guard ring [10]. Xue Li et al. analyzed the electrical crosstalk of planar-type 32 × 32 30 μm and 512×1 25 μm InGaAs linear detectors with the laser-beam-induced current technique (LBIC). The guard ring reduced the electrical crosstalk to 2.5% [11]. However, all above studies result in the great reduction in the fill factor and the performance degradation of InGaAs FPA detectors. Therefore, some other methods are desired to suppress the electrical crosstalk of high–density FPA detectors.

Owing to materials defects and imperfect fabrication processing, there are some inoperable pixels in planar-type InGaAs detectors. The photogenerated carriers of the inoperable pixels cannot be conducted to the corresponding readout element of the readout integrated circuit (ROIC) but diffuse laterally to the adjacent pixels, resulting in critical electrical crosstalk, which can be observed and determined from the response gray value change of the pixels. In this paper, a guard-hole structure is proposed and designed to suppress the electrical crosstalk generated by the inoperable pixel in a 640 × 512 15 μm InGaAs infrared detector. Furthermore, a guard-ring structure is designed and prepared with the same process for comparison. The test results show that the guard hole effectively suppresses electrical crosstalk, and the detector with a guard hole has a higher quantum efficiency (QE) and detectivity, as well as a lower dark current compared with the detector with a guard ring.

## 2. Fabrication of FPAs

The p-type doping-intrinsic-n-type-doping (PIN) InGaAs/InP epitaxial material comprises a 1 μm N-InP top layer, a 2.8 μm InGaAs absorbing layer, a 0.4 μm N-InP buffer layer with a doping concentration of 2 × 10^18^ cm^–3^, and a 625 μm InP substrate layer. A SiN layer is deposited on the EPI wafer by PECVD after pre-cleaning with 1% HCl, then the Zn diffusion hole is fabricated with SiN dry and wet etching. The guard-hole structure has four guard holes at the four corners of every pixel; the diameter of the guard hole is 2 μm, as shown in Figure 1a. The guard-ring structure has the guard ring around a pixel; the width of the guard ring is 2 μm, and the length is 15 μm, as shown in Figure 1b. Zn diffusion is carried out with Zn_3_P_2_ in a sealed tube. After that, the second SiN layer is deposited on the wafer for Zn activation anneal and SiN is dry-etched with RIE. P-type-metal is evaporated with an E-beam. As shown in Figure 1c,d. Then, the InP and InGaAs around the active region were etched to expose the n^+^-InP buffer layer with wet etching, and the n-type metal is evaporated with an E-beam. The P-type metal and n-type metal are annealed with RTA for ohmic contact. The connecting metal is deposited with an E-beam, and the guard hole and the guard ring are connected to the n-type metal with a connecting metal. An indium bump was deposited after polishing the substrate and the antireflection coating deposition. Finally, the InGaAs photodiode arrays (PDAs) were hybrid-integrated with the Complementary Metal Oxide Semiconductor (CMOS) readout integrated circuit (ROIC) with the flip–chip bonding process.

## 3. Results and Discussion

Both types of FPA detectors were measured with a Pulse Instruments 7700 (Torrance, CA, USA) FPA test system and EMVA 1288 (European Machine Vision Association, Barcelona, Spain) standard at room temperature, the light source was a 1550 nm LED with a light intensity of 0.55 μW/cm^2^, and the integration time was 6 ms. We obtained the response gray values of pixels, dark current density, QE, voltage responsivity, and detectivity, respectively.

### 3.1. Electrical Crosstalk

The electrical crosstalk effect can be evaluated by the response gray value of a pixel. There are some inoperable pixels in 640 × 512 15 μm InGaAs FPA without a guard structure, as shown in Figure 2a. The response gray value of an inoperable pixel and the pixels around it are shown in Figure 2b. The center black square is the response gray value R_c_ of an inoperable pixel, and it reaches the saturated value of the FPA. The average response gray value R_f_ of four adjacent pixels is 24,757, and the average response gray value R_a_ of 640 × 512 pixels is 28,583. The difference between R_f_ and R_a_ is 3826. To quantitatively express the electrical crosstalk, the calculation expression of crosstalk C is given by Equation (1). The electrical crosstalk C generated by the center inoperable pixel to the adjacent pixels is up to 13.4%. One of the main reasons for the occurrence of the crosstalk is shown in Figure 2c, which displays the middle pixel becoming an inoperable pixel because of the insufficient height of the indium bump. The photocurrent of the inoperable pixel diffuses laterally and is collected by the four adjacent pixels under reverse bias, resulting in serious electrical crosstalk.

Figure 3a shows the response grayscale of the detector with a guard hole. The response gray value of the four adjacent pixels R_f_ is 1425 lower than the average value of the detector R_a_. The response grayscale of the detector with a guard ring is shown in Figure 3b. The R_f_ is 119 lower than R_a_, as shown in Table 1.
(1)$$C=\frac{\left|R_{f}-R_{a}\right|}{R_{a}}\times 100$$

The calculation results show that the electrical crosstalk of the detector with a guard ring is 0.40%, and the crosstalk of the detector with a guard hole is 4.5%. The electrical crosstalk is almost eliminated by the guard-ring structure. The guard-hole structure effectively suppresses the electrical crosstalk to be 66% lower than that of the detector without a guard structure. The guard ring behaves like a closed channel, and the photocurrent of the inoperable pixel is collected completely by the guard ring and conducted to the n electrode, as shown in Figure 4a. The guard holes behave like four wells that collect the photocurrent from the inoperable pixel, but there is a gap between the two guard holes, and part of the photocurrent diffuses to adjacent pixels and is collected by the adjacent four pixels through the gaps, resulting in a small amount of electrical crosstalk, as shown in Figure 4b. Therefore, the guard-ring structure is the most effective structure for suppressing the electrical crosstalk of InGaAs FPAs.

### 3.2. Dark Current Density

The dark current density J_d_^ring^ of the detector with a guard ring is higher than the J_d_^hole^ of the detector with a guard hole, as shown in Figure 5. The p-n junction area A_pn_^ring^ of the guard ring is bigger than the A_pn_^hole^ of the guard hole. According to the calculation, the A_pn_^hole^ is 35.33 μm^2^, and the A_pn_^ring^ is 84.26 μm^2^. The value of the A_pn_^ring^/A_pn_^hole^ is 2.38, while the ratio of the J_d_^ring^ to the J_d_^hole^ is 2.52. The ratio of the A_pn_^ring^/A_pn_^hole^ is in good agreement with the ratio of the J_d_^ring^/J_d_^hole^, as shown in Table 2. It can be seen that the increase in the current density is caused by the increase in the p-n junction area.

### 3.3. Quantum Efficiency (QE)

QE is a key parameter of InGaAs detectors. The QEs of the detectors with a guard ring and guard hole are 28.64% and 47.14%, respectively. The QE^hole^ of the detector with a guard hole is 1.65 times that of the detector with a guard ring. The area of the guard hole is 7.056 μm^2^, while the area of the guard ring is 56 μm^2^. The fill factor of the detector with a guard hole, F^hole^, is 96.9%, while the fill factor of the detector with a guard ring, F^ring^, is 75.1%. Under reverse bias, the photocurrent generated by the photo-sensitive area is collected by the guard hole and the guard ring and conducted to the n electrode, respectively, resulting in photocurrent loss. According to Equations (2) and (3), the loss of the photocurrent leads to a reduction in the responsivity R and QE. Because the fill factor of the detector with a guard ring is lower than that of the detector with a guard hole, the loss of the photocurrent caused by the guard ring is greater than that of the guard hole. Therefore, the QE of the detector with a guard ring is lower than that of the detector with a guard hole.
(2)$$\eta =\frac{I_{p}h\nu }{\mathrm{qP}}=R\frac{h\nu }{q}$$
(3)$$R=\frac{I_{p}}{P}$$

In Equations (2) and (3), η is QE, I_p_ is the photocurrent, h is the Planck constant, ν is the frequency of light, q is the electronic quantity, p is the optical power, and R is the current responsivity.

### 3.4. Detectivity D*

The detectivity of the detector with a guard hole, D*_hole_, is 2.89 × 10^12^ cm·Hz^1/2^/W, and the detectivity of the detector with a guard ring, D*_ring_, is 1.22 × 10^12^ cm·Hz^1/2^/W, as shown in Table 3.

The detectivity is determined by the voltage responsivity R_v_ and the noise voltage V_n_, as shown in Equation (4). Since the F^ring^ is lower than the F^hole^, the R_v_^ring^ is lower than the R_v_^hole^, and the value of the R_v_^hole^/R_v_^ring^ is 1.65. At the same time, the A_pn_^ring^ is bigger than the A_pn_^hole^, causing the J_d_^ring^ and the V_n_^ring^ higher than the J_d_^hole^ and the V_n_^hole^; the ratio of the V_n_^hole^ to the V_n_^ring^ is 0.7. D* and is proportional to (R_v_/V_n_) from Equation (4). Hence, It can be calculated and obtained that the D*_hole_ is 2.36 times the D*_ring_, which is well consistent with the measured detectivity ratio of the D*_hole_/D*_ring_ = 2.37, as shown in Table 3.
(4)$$D^{*}=\frac{R_{v}}{V_{n}}\sqrt{A_{d}\Delta f}$$

In Equation (4), R_v_ is voltage responsivity, V_n_ is noise voltage, A_d_ is the area of a pixel, and Δf is the noise equivalent bandwidth.

## 4. Conclusions

In summary, the suppression of electrical crosstalk caused by an inoperable pixel is studied by fabricating the guard hole and guard ring in a 640 × 512 15 μm InGaAs infrared detector and utilizing the method of the response gray value difference between pixels. The results show that the electrical crosstalk values of the detectors with a guard ring and a guard hole are 0.4% and 4.5%, respectively. However, because the p-n junction area of the guard ring is larger than that of the guard hole, the results show the dark current with a guard ring is 2.52 times that of with a guard hole. In addition, the fill factor with a guard ring is lower than that of with a guard hole, and the QE with a guard hole is 1.65 times that of with a guard hole. The two factors determined that the detectivity with a guard hole is 2.36 times that of with a guard ring. Therefore, the detector with a guard hole not only effectively suppresses the electrical crosstalk but also has a higher photoelectric performance.

## Figures and Tables

**Figure 1 micromachines-13-01797-f001:**
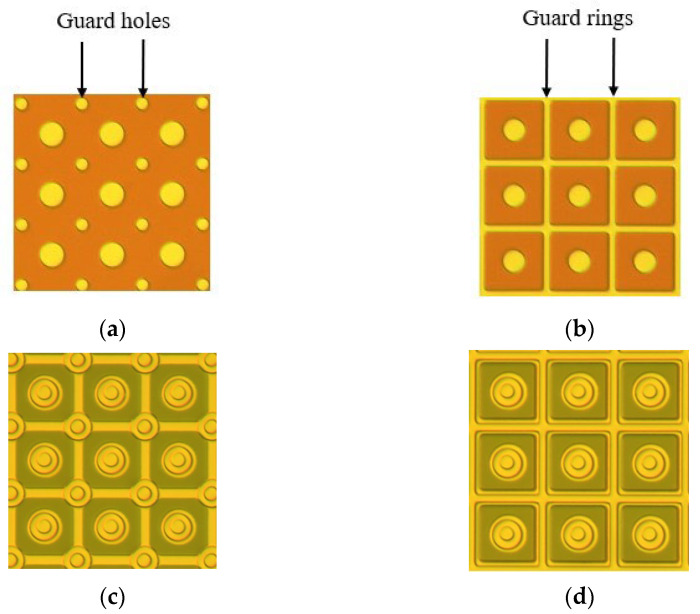
(**a**) After the SiN etching of the guard-hole structure. (**b**) After the SiN etching of the guard-ring structure. (**c**) After the P-metal deposition of the guard-hole structure. (**d**) After the P-metal deposition of the guard-ring structure.

**Figure 2 micromachines-13-01797-f002:**
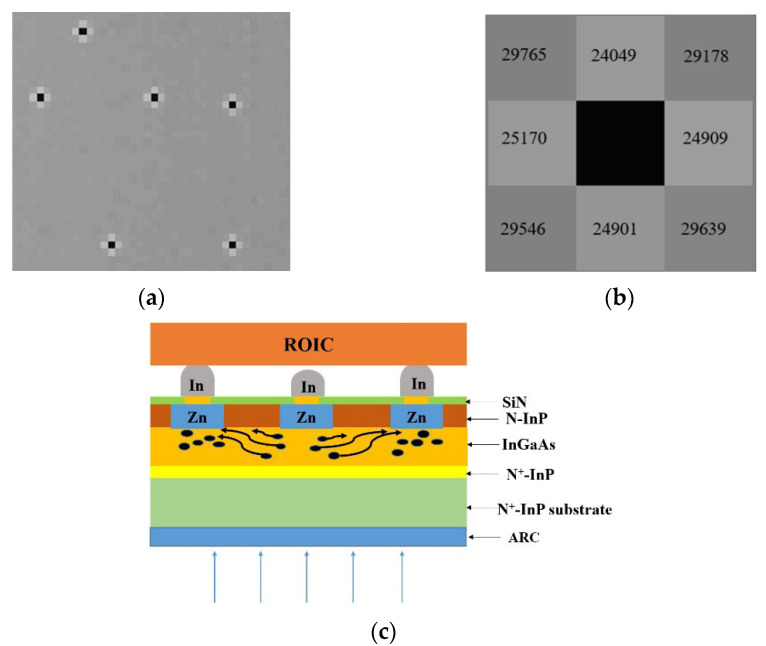
(**a**) scatter diagram of Inoperable pixels. (**b**) Response gray value of electrical crosstalk of inoperable pixel. (**c**) The schematic diagram of electrical crosstalk of inoperable pixel.

**Figure 3 micromachines-13-01797-f003:**
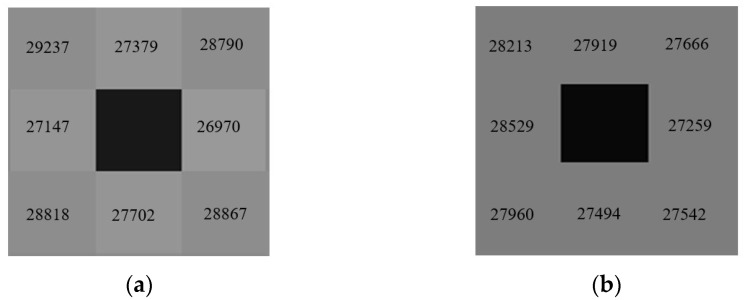
(**a**) Response grayscale of the detector with a guard hole; (**b**) Response grayscale of the detector with a guard ring.

**Figure 4 micromachines-13-01797-f004:**
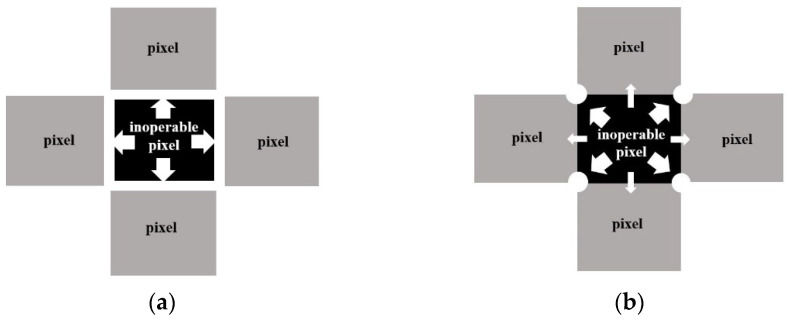
Schematic of suppression effect of electrical crosstalk of two structures. (**a**) Guard-ring structure. (**b**) Guard-hole structure.

**Figure 5 micromachines-13-01797-f005:**
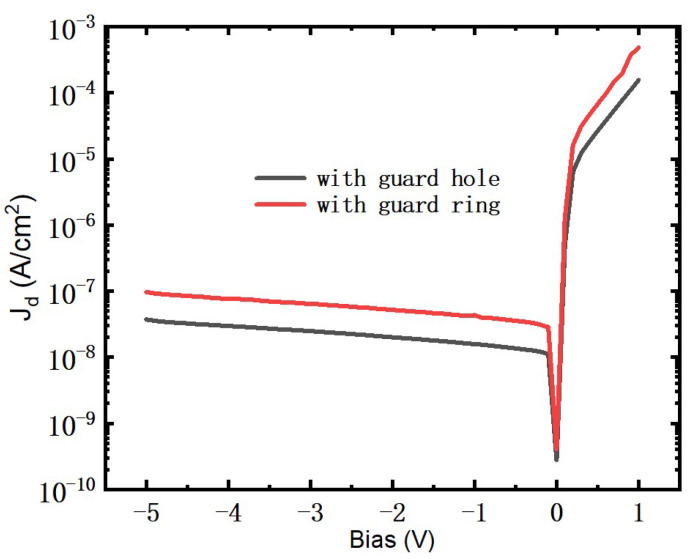
Dark current density of two structures.

**Table 1 micromachines-13-01797-t001:** The average response gray value.

Parameter	R	Ra	|R_f_ − R_a_|	C
Guard ring	27,800	27,919	119	0.40%
Guard hole	27,299	28,724	1425	4.5%

**Table 2 micromachines-13-01797-t002:** The ratio of A_pn_, J_d_ between the guard-hole and the guard-ring structure.

Parameter	A_pn_ /μm^2^	J_d_ /nA/cm^2^
Guard ring	84.26	28.5
Guard hole	35.33	11.2
Ratio of guard ring to guard hole	2.38	2.52

**Table 3 micromachines-13-01797-t003:** The ratio of Rv, Vn, D, and D* between the guard-hole and guard-ring structure.

Parameters	R_v_ *v*/*w*	V_n_ v	D* cm·Hz^1/2^/w
Guard hole	5.89 × 10^11^	2.42 × 10^−3^	2.89 × 10^12^
Guard ring	3.56 × 10^11^	3.46 × 10^−3^	1.22 × 10^12^
Ratio of guard ring to guard hole	1.65	0.7	2.37

## Data Availability

Data are contained within the article.

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
