# Peer review of "Suppression of the Electrical Crosstalk of Planar-Type High-Density InGaAs Detectors with a Guard Hole"

_micromachines, 2022, doi:10.3390/mi13101797_

Round 1

Reviewer 1 Report

The problem of electrical crosstalk is crucial for high density IR FPAs. This still true for InGaAs SWIR technology and it is very important to address it. The paper is well written and very interesting for the IR detector community. However, additions are necessary to improve the scientific quality of the paper before publication.

- the end of the abstract, reporting the results obtained on figure of merit with guard ring or guard hole has to be re-written in order to be more condensed, concise.

- I have some problems to consider figures in a introduction (Fig. 1 and 2). Please, try to be more general in the introduction in order to avoid the figures. Moreover, InGaAs FPA are commercially available. The authors have to discuss about the problem of electrical crosstalk on currently commercial devices available.

- compare to guard ring, guard hole suppress electrical crosstalk inducing better QE, detectivity and lower dark current. Is it possible to quantify such advantage by a specific measurement ? I think about FTM measurements on FPA. Discussion about that by the authors would be welcome.

- if electrical crosstalk between pixels is present, the size of pixel is not well know. So, how to calculate, evaluate the real quantum efficiency or dark current density of the device ? This point is very important, because many quantitative comparisons between the two designs have been given to confirm the advantage of guard hole.

Author Response

Point 1: The end of the abstract, reporting the results obtained on figure of merit with guard ring or guard hole has to be re-written in order to be more condensed, concise.

Response 1we have re-written the end of the abstract as follows:

Experimental results show the detector with guard-hole structure has higher performance compared with the detector with guard-ring structure, the dark current density is reduced by 60%, the QE is increased by 64.5%, and the detectivity is increased by1.36 times, respectively. The test results indicate that the guard-hole structure provides a novel suppression method for electrical crosstalk with higher performance of the FPA.

Point 2: I have some problems to consider figures in a introduction (Fig. 1 and 2). Please, try to be more general in the introduction in order to avoid the figures. Moreover, InGaAs FPA are commercially available. The authors have to discuss about the problem of electrical crosstalk on currently commercial devices available.

Response 2I have moved paragraph 2, Fig.1and Fig.2 in Introduction to section 3.1 for explaining the electrical crosstalk in InGaAs FPA without guard structure.

Moreover, I added the problem of electrical crosstalk on currently commercial devices available in Introduction as follows:

 Many industrial, scientific and commercial applications rely on accurate knowledge of the quantitative information in image data, including color balance correction in imaging cameras, shape-measurement in industrial robotic and quality-control applications. These applications put forward higher requirements for reducing the electrical crosstalk of InGaAs detectors.

Point 3: compare to guard ring, guard hole suppresses electrical crosstalk inducing better QE, detectivity and lower dark current. Is it possible to quantify such advantage by a specific measurement? I think about FTM measurements on FPA. Discussion about that by the authors would be welcome.

Response 3: Yes, I agree with you very much that FTM is an effective way to measure electrical crosstalk. Unfortunately, we have no relevant equipment for FTM test and We could not find the relevant organization for FTM test soon because of COVID-19. we test FPA performance with uniform parallel light. electrical crosstalk occurs between adjacent pixels in planar-type InGaAs detectors. However, the electrical crosstalk generated by inoperable pixel is the most serious, as shown in the following figure.

The photocurrent of inoperable pixel diffuses laterally and is collected by the four adjacent pixels under reverse bias. Therefore, we take the electrical crosstalk generated by inoperable pixel as the object to study the suppression of electrical crosstalk with guard hole and guard ring. This method is also an efficient way to indicate the reduced crosstalk.

Point 4: if electrical crosstalk between pixels is present, the size of pixel is not well known. So, how to calculate, evaluate the real quantum efficiency or dark current density of the device? This point is very important, because many quantitative comparisons between the two designs have been given to confirm the advantage of guard hole.

Response 4: The pitch of the planar-type InGaAs FPA is 15μm, and the guard hole or the guard ring is part of a pixel. So, the size of pixel is 15×15 μm2, all results are calculated based on 15 ×15 μm2. Because junction area of guard hole and guard ring increase the junction area of pixel, which cause dark current increase. Due to guard hole and guard ring connecting to n electrode, resulting in loss of photocurrent of pixel, which leads to the decrease of quantum efficiency. Junction area of guard hole is smaller than that of guard ring, compare with guard ring,the structure with guard hole has advantage. All the test results indicate this point.

Reviewer 2 Report

This manuscript demonstrated an InGaAs FPA with guard hole structures and compared the performances to the FPA with guard ring structures. The authors claim that the FPA with guard hole structures induces reduced electrical crosstalk, decreased dark current as well as increased quantum efficiency and detectivity.

First of all, this manuscript is focused on the structure of optoelectronic devices based on III-V materials. The research content does not match the interested area of ‘micromachines’.

In addition, the manuscript is lack of necessary information and discussion.

1. The review of literatures on the research of electrical crosstalk is not complete in the introduction part. In fact, many kinds of structures have reported to restrain the electrical crosstalk of FPAs. Also, different characterization methods of electrical crosstalk exist.

2. The experimental process needs to be more detailed. The material parameters of some layers are lack in the manuscript. The FPA structures and fabrication process are also too brief, e.g. if passivation layer was contained, which deposition processing was applied. Especially, the detailed dimension of the guard holes and guard rings should be marked, otherwise, the area calculations in the dark current and QE part are not clear for the readers.

3. The most direct and efficient way to indicate the reduced crosstalk should use the laser spot with small size. Therefore, the authors are suggested to add the direct way for crosstalk measurements.

4. It is strange that the junction area is reduced for the structure with guard holes. From the schematic structures, it seems that the structure with guard holes have larger optical sensitive area.

Author Response

First of all, this manuscript is focused on the structure of optoelectronic devices based on III-V materials. The research content does not match the interested area of ‘micromachines’.

Response: This manuscript is submitted to Special issue “Terahertz and Infrared Metamaterial Devices” of micromachines. We think it match the micromachines.

Point 1: The review of literatures on the research of electrical crosstalk is not complete in the introduction part. In fact, many kinds of structures have reported to restrain the electrical crosstalk of FPAs. Also, different characterization methods of electrical crosstalk exist.

Response 1: We added the different methods and some references about electronical crosstalk as follows:

Estribeau, M.et al masked active area to reduce crosstalk by recovering a symmetry and without an important loss in quantum efficiency of CMOS imager sensor with Modulation Transfer Function (MTF) [7].

[3] Seshadri, S.; Cole, D. M.; Hancock, B. R.; & Smith, R. M.  Mapping electrical crosstalk in pixelated sensor arrays[J]. High Energy, Optical, and Infrared Detectors for Astronomy III, 2008, 7021.

[4] Sanders, T. J.; Caraway, E. L., Hess, G. T., Newsome, G. W., Fischer, T. Modeling and test of pixel cross-talk in HgCdTe focal plane arrays[J]. Proceedings of Spie the International Society for Optical Engineering, 2001, 4369:458-466.

[5] Musca, C.A.; Dell, J.M.; Faraone, L; Bajaj, J.; Pepper, T.; Spariosu, K.; Blackwell, J.; Bruce, C. Analysis of crosstalk in HgCdTe p-on-n heterojunction photovoltaic infrared sensing arrays. J. Electron. Mater. 28, 617–623 (1999).

[6] Zhu, Y. M.; Li X.; Wei J.; Li, J.W.; Gong, H. M. Analysis of cross talk in high density mesa linear InGaAs detector arrays using tiny light dot[J]. Proc Spie, 2012, 8419(5):260-264.

[7] Estribeau M.; Magnan P. Pixel Crosstalk and Correlation with Modulation Transfer Function of CMOS Image Sensor[J]. Proceedings of SPIE - The International Society for Optical Engineering, 2005, 5677:98-108.

[8] Chamberlain, S. G.; Harper, H D. MTF simulation including transmittance effects and experimental results of charge-coupled imagers[J]. Electron Devices IEEE Transactions on, 1978, ED-25:145-154.

Point 2: The experimental process needs to be more detailed. The material parameters of some layers are lack in the manuscript. The FPA structures and fabrication process are also too brief, e.g. if passivation layer was contained, which deposition processing was applied. Especially, the detailed dimension of the guard holes and guard rings should be marked, otherwise, the area calculations in the dark current and QE part are not clear for the readers.

Response 2: We added the process details as follows:

 A SiN layer was deposited on the EPI wafer by PECVD after pre-clean with 1% HCl, then the Zn diffusion hole was fabricated with SiN dry and wet etch. the diameter of guard hole is 2 μm, the width of guard ring is 2 μm, and the length is 15 μm. Zn diffusion is carried out with Zn3P2 in sealed tube. After that, the second SiN layer is deposited on the wafer for Zn activation anneal and SiN is dry-etched with RIE. P-type-metal is evaporated with E-beam. The InP and InGaAs around the active region was etched to expose n+-InP buffer layer with wet etching, and n-type metal is evaporated with E-beam. P-type metal and n-type metal is annealed with RTA for ohmic contact. The connecting metal is deposited with E-beam.

Point 3: The most direct and efficient way to indicate the reduced crosstalk should use the laser spot with small size. Therefore, the authors are suggested to add the direct way for crosstalk measurements.

Response 3: Yes, what you suggest is a good way to measure the crosstalk. Unfortunately, we do not have the related equipment to measure it. We could not find the relevant organization for FTM test soon because of COVID-19. In addition, we test FPA performance with uniform parallel light.   electrical crosstalk occurs between adjacent pixels in planar-type InGaAs detectors. However, the electrical crosstalk generated by inoperable pixel is the most serious, as shown in figure 2a

The photo current of inoperable pixel diffuses laterally and is collected by the four adjacent pixels under reverse bias. As shown in figure 2c.

Therefore, we take the electrical crosstalk generated by inoperable pixel as the object to study the suppression of electrical crosstalk with guard hole and guard ring. This method is also an efficient way to indicate the reduced crosstalk. Besides, Compared with laser spot method, our research method is closer to the actual application scene.

Point 4: It is strange that the junction area is reduced for the structure with guard holes. From the schematic structures, it seems that the structure with guard holes have larger optical sensitive area.

Response 4: From the schematic structures, the guard holes increase the optical sensitive area of the pixels, but photocurrent generated by optical sensitive area of the guard holes is conducted to n electrode, resulting in loss of photocurrent. Therefore, actually, the effective optical sensitive area of the structure with guard holes is reduced. But the schematic structure of guard holes is a little bigger,I change a new one with smaller guard holes, as shown in figure 4b.

Round 2

Reviewer 1 Report

Authors have taken into account comments and remaks made by the referees. 

Reviewer 2 Report

The authors have addressed all the previous concerns of the reviewers.